# Secreted MbovP0145 Promotes IL-8 Expression through Its Interactive β-Actin and MAPK Activation and Contributes to Neutrophil Migration

**DOI:** 10.3390/pathogens10121628

**Published:** 2021-12-15

**Authors:** Doukun Lu, Hui Zhang, Yiqiu Zhang, Gang Zhao, Farhan Anwar Khan, Yingyu Chen, Changmin Hu, Liguo Yang, Huanchun Chen, Aizhen Guo

**Affiliations:** 1The State Key Laboratory of Agricultural Microbiology, Huazhong Agricultural University, Wuhan 430070, China; doukunlu@webmail.hzau.edu.cn (D.L.); dkyzhanghui@163.com (H.Z.); zyq199709@163.com (Y.Z.); zhaog0524@163.com (G.Z.); chenyingyu@mail.hzau.edu.cn (Y.C.); hcm@mail.hzau.edu.cn (C.H.); chenhch@mail.hzau.edu.cn (H.C.); 2College of Veterinary Medicine, Huazhong Agricultural University, Wuhan 430070, China; 3Department of Animal Health, The University of Agriculture, Peshawar 25120, Pakistan; fahan82@aup.edu.pk; 4College of Animal Science and Technology, Huazhong Agricultural University, Wuhan 430070, China; liguoyang2006@163.com; 5Key Laboratory of Development of Veterinary Diagnostic Products, Ministry of Agriculture, Huazhong Agricultural University, Wuhan 430070, China; 6Hubei International Scientific and Technological Cooperation Base of Veterinary Epidemiology, Key Laboratory of Preventive Veterinary Medicine in Hubei Province, Wuhan 430070, China

**Keywords:** *Mycoplasma bovis*, secreted proteins, Mbov_0145, β-actin, IL-8

## Abstract

*Mycoplasma bovis* (*M. bovis*) is an important pathogen of cattle responsible for huge economic losses in the dairy and beef industries worldwide. The proteins secreted by *M. bovis* are mainly related to its adhesion, invasion, virulence, and intracellular survival and play a role in mycoplasma–host interactions. In our previous study, we found MbovP0145, a secreted protein present in the *M. bovis* secretome, but little is known about its function. In this study, we assessed the inflammatory characteristics and underlined mechanism of this inflammation of recombinant MbovP0145 (rMbovP0145). For this, bovine lung epithelial cells (EBL) were stimulated by rMbovP0145 to see the IL-8 production in a time- and dose-dependent manner. We observed that rMbovP0145 increased the production of IL-8 via ERK1/2 and P38 pathway activation. Further, the effect of the *M. bovis* ΔMbov_0145 mutant and its complementary strain on IL-8 mRNA expression was also confirmed. A pulldown assay of the GST-tagged MbovP0145 protein with mass spectrometry demonstrated that β-actin could specifically interact with rMbovP0145 to mediate the IL-8 signaling. As knockdown of β-actin expression with RNA interference in EBL cells decreased the mRNA expression of IL-8 and the phosphorylated ERK1/2 and P38 proteins, whereas disrupted actin polymerization by cytochalasin D led to a significantly higher IL-8 expression and MAPK phosphorylation in rMbovP0145-stimulated cells. Compared to *M. bovis* HB0801 and its complementary strain, the culture supernatant of EBL cells infected with the *M. bovis* ΔMbov_0145 mutant induced less neutrophil migration to the lower chamber in a transwell system. In conclusion, MbovP0145 promoted IL-8 expression by interacting with β-actin through activation of the MAPK pathway, thus contributing to neutrophil migration.

## 1. Introduction

Mycoplasmas belong to the class Mollicutes, and many are highly host-specific and successful pathogens [1]. Several characteristics, such as the small size of genomes, the absence of cell walls, the limited metabolic pathways, and the shortage of conventional secretion systems and toxins, make them different from other microbes regarding pathogenesis [2]. *Mycoplasma bovis* (*M. bovis*) is an important bovine pathogen and causes different diseases with variety of clinical manifestations, including pneumonia, mastitis, polyarthritis, otitis media, and genital disorders [3]. Since the first report on its association with mastitis and respiratory diseases in cattle were reported in 1961 and 1976, respectively [4]. Now, it has become a worldwide problem for the beef and dairy industries. The pathogenesis of *M. bovis* can be attributed to a variety of membrane proteins related to virulence. These include adhesins, a family of variable surface proteins (VSPs) modifying as antigen diversity; proteins with nuclease activities [5], IgG-binding/cleavage abilities [6], and the ability to form biofilms [7]; and cytoplasm proteins, such as MbovP327 and MbovP328, with phosphodiesterase abilities to degrade cyclic dinucleotides and/or nanoRNAs, which is essential to *M. bovis* growth under cocultures with host cells [8].

Microbial secreted proteins, such as exotoxins, are generally considered the key factors driving pathogenesis [9]. So far, only a few secreted proteins have been thoroughly investigated. One such protein is MbovP280, which induces apoptosis by binding to the anti-apoptosis regulator αB-crystallin (CRYAB) [10]. MBOV_RS02825 encodes a secretory nuclease and is able to degrade a DNA matrix of neutrophil extracellular traps (NETs) [11]. Our previous research on the secretome of *M. bovis* has found that *M. bovis* has over 178 secreted proteins from *M. bovis* HB0801 P1 and its attenuated strain P150 [12]. Therefore, there is an urgent need to explore the biological functions of more secreted proteins and uncover their role in the pathogenesis of *M. bovis*. 

Within the *M. bovis* secretome, we previously identified a putative lipoprotein MbovP0145 that was upregulated in a virulent *M. bovis* HB0801 strain compared with its attenuated strain P150 [12]. It is also a potential serological marker of *M. bovis* infection, which is conserved in the genome of most *M. bovis* clinical isolate strains [12]. In addition, based on its sequence characteristic, it is homologous to the Bacteroides surface protein A (BspA) family, which has been reported to be involved in bacterial adhesion, invasion, and triggering the release of proinflammatory cytokines [13]. Therefore, this study aimed to investigate that the role of MbovP0145 in *M. bovis* infection and to elucidate the underlying mechanism. Our findings demonstrate that MbovP0145 can induce IL-8 expression in embryonic bovine lung (EBL) cells by interacting with β-actin and activating the mitogen-activated protein kinase (MAPK) pathway. 

## 2. Materials and Methods

### 2.1. In Silico Analyses

In silico analyses of Mbov_0145 CDS were carried out using the National Center for Biotechnology Information (NCBI) database. The coding domain sequence of Mbov_0145 from the *M. bovis* HB0801 genome (position nt162942-nt164387) was retrieved from the NCBI database (GenBank accession no. NC_018077.1). Analyses of codon usage, codon optimization, and hydropathy were carried out using ProtParam (Expsy). Cellular localization and signal peptide prediction were performed using PSORTb and SignalP 5.0, respectively. The amino acid sequences of MbovP0145 and orthologs were obtained from the NCBI database, and the Multiple Sequence Comparison by Log-Expectation (MUSCLE) was used for multiple protein sequence alignment. A phylogenetic tree was constructed using the neighbor-joining method with 1000 nonparametric bootstrap replicates in MEGA-X.

### 2.2. Cultivation of Bacterial Strains and Cell Lines

The *M. bovis* WT strain HB0801 (China Center for Type Culture Collection; CCTCC No. M2010040) was isolated from the lung of infected beef calf in Hubei Province, China by our laboratory in 2008 [14]. An Mbov_0145 mutant *M. bovis* ΔMbov_0145 designed as T6.93 (CCTCC No. M2020081) was identified from an *M. bovis* transposon mutant library previously generated by a mini-Tn4001 in this lab [15]. The transposon insertion site was determined using Tn sequencing. All mycoplasma strains were cultivated to the mid-logarithmic (log) phase in pleuropneumonia-like organism (PPLO) medium (BD Company, Sparks, MD, USA) as previously described [16] with the addition of 100 μg/mL gentamicin for mutant growth. The *Escherichia coli* (*E. coli*) strains DH5α and BL21 (TransGen Biotech, Beijing, China) were grown in Luria–Bertani (LB) medium with antibiotics when necessary and used in DNA cloning and protein expression.

The EBL cell line (kindly provided by Dr. Eric Baranowski from Université de Toulouse) was grown in Dulbecco’s modified Eagle’s medium (DMEM)-based medium consisting of DMEM (Gibco, Waltgam, MA, USA) supplemented with 1% nonessential amino acids (NEAA; Gibco) and 10% heat-inactivated fetal bovine serum (FBS; Gibco) at 37 °C and 5% CO_2_. The HEK293T cell line was purchased from CCTCC and cultured in DMEM (Gibco) supplemented with 10% heat-inactivated FBS (Gibco).

### 2.3. Cloning, Expression, and Purification of a Recombinant MbovP0145 Protein

After codon adaptation for expression in Mbov_0145 genes in *E. coli*, the full length of the modification Mbov_0145 gene was cloned into the pET-30a, pGEX-6p-1, pCAGGS-HA, and pEGFP-C1 vectors, respectively. The plasmids pCAGGS-HA-Mbov_0145 and pEGFP-C1-Mbov_0145 were extracted using an endo-free plasmid kit (Omega Bio-tek, GA, USA). The recombinant plasmids pET-30a-Mbov_0145 and pGEX-6p-Mbov_0145 were then transformed into *E. coli* BL21 competent cells (Vazyme, Nanjing, China) to express the recombinant fusion proteins His-rMbovP0145 and GST-rMbovP0145. The His-rMboP0145 protein was induced by 1 mM isopropyl-β-D-thiogalactopyranoside-de (IPTG) and purified by a high-affinity Ni-NTA resin column (Thermo Fisher Scientific, Rockford, IL, USA) and ultrafiltration in phosphate-buffered saline (PBS). GST-rMbovP0145 was purified by affinity chromatography with glutathione–Sepharose 4B beads (GE Healthcare, Piscataway, NJ, USA). Before using the recombinant proteins to stimulate cells, repeated treatments with Triton X-114 were carried out to remove the endotoxin (<0.001-EU/μg protein) as described previously [17]. Proteins were quantified using a bicinchoninic acid (BCA) protein assay kit (Beyotime Biotechnology, Shanghai, China). The mouse antiserum to rMbovP0145 was prepared previously, and its immunity serum titer was confirmed by iELISA [12]. 

### 2.4. Construction of Strain Complementary to M. bovis ΔMbov_0145

#### 2.4.1. In Trans Complementation of *M. bovis* T6.93 via a Plasmid Carrying the Gene Mbov_0145

To construct the strain complementary to the mutant T6.93 (*M. bovis* ΔMbov_0145), the entire coding region with *M. agalactiae* P40 promoter and the full length of Mbov_0145 were amplified by PCR; the primers are presented in Table 1 and were integrated by overlapped extension PCR. After digestion with restriction enzyme Not I, the amplified fragment was ligated into the pOH/P plasmid, thus generating the recombinant plasmid pOH/P–Mbov_0145. After the sequence insertion was confirmed to be correct, the plasmid was transferred into T6.93 by PEG8000, yielding the complementary strain designated as CT6.93. The T6.93 and CT6.93 strains were cultured in a medium containing 100 μg/mL gentamycin and 10 μg/mL puromycin, respectively, and their growth curves were determined with the plate colony-counting method. 

#### 2.4.2. Western Blot Analysis of MbovP0145 Expression in *M. bovis* Strains

Whole-cell lysate (WCL) was extracted from the strains *M. bovis* HB0801, mutant T6.93, and complement strain CT6.93, which were grown to the mid-log phase in PPLO medium at 37 °C, as described above. An equal amount of the WCL from each strain was separated on 12% SDS-PAGE gel and transferred onto polyvinylidene fluoride membranes. Immunoblots were then performed with a 1:500 dilution of the mouse anti-rMbovP0145 serum or anti-MbovP0579 serum previously prepared by this laboratory as the control of the membrane protein [18] to detect MbovP0145 expression in *M. bovis* strains.

### 2.5. EBL Cells Either Infected with M. bovis or Treated with rMbovP0145 

For *M. bovis* infection, EBL cells were seeded in 12-well plates at a concentration of 2 × 10^5^ cells/well and incubated at 37 °C overnight prior to infection. Next, a 1 mL culture of each strain was centrifuged (10,000× *g* for 5 min), washed with sterile PBS three times, and resuspended in fresh DMEM of the same initial volume. The cells were then washed three times with PBS and infected with *M. bovis* suspension at a multiplicity of infection of 1000 for 6 h, 12 h, and 24 h. For the protein treatment, the monolayer of the EBL cells was treated with rMbovP0145 either at different concentrations (2 μg/mL, 4 μg/mL, 6 μg/mL, 8 μg/mL, 10 μg/mL, and 20 μg/mL) in complete medium for 12 h or at the optimum working concentration (8 μg/mL) in complete medium for various times until harvest. Meanwhile, the EBL cells were mock-treated with PBS as the negative control and treated with *M. bovis* and LPS as the positive control.

### 2.6. Analysis of IL-8 mRNA Expression with Quantitative Real-Time PCR 

Total RNA was extracted from infected EBL cells using the TRIzol reagent (Invitrogen, Carlsbad, CA, USA) according to the manufacturer’s instructions, and the RNA concentration was assessed using NanoDrop 2000 (Thermo Fisher Scientific). Next, the RNA was reverse-transcribed into cDNA using a HiScript II Q Select RT SuperMix kit (Vazyme) according to the manufacturer’s instructions. Each cDNA sample was analyzed in triplicate using qPCR. PCR amplification was performed using the ABI ViiA 7 Real-Time PCR system (Applied Biosystems, Carlsbad, CA, USA) by using GAPDH as the internal control. The primer sequences are presented (Table 1). The relative gene expression levels were determined by referring to the 2^−ΔΔCT^ calculation method. Each treatment was carried out in three repeats, and all experiments were performed independently three times. 

### 2.7. GST Pull-Down Assay to Identify Interactive Protein of MbovP0145

EBL cells were harvested by centrifugation at 300× *g* for 10 min at 4 °C after washing three times with PBS. The pellet was suspended in 1 mL NP40 lysis buffer (Beyotime Biotechnology) containing 1 mM PMSF for 30 min. After centrifugation at 10,000× *g* for 15 min, the total cell supernatant was harvested. Next, 500 µL of the cell supernatant was preincubated with 50 µL glutathione–Sepharose beads for 2 h and then centrifugated at 300× *g* for 5 min at 4 °C to remove the nonspecific proteins. The glutathione–Sepharose beads were conjugated to either GST or GST-rMbovP0145 proteins at 4 °C for 4 h, washed twice with NP40 buffer, and incubated with the supernatant at 4 °C overnight. The beads were then washed four times with ice-cold NP40 buffer by a rocking–turn blending, followed by elution with SDS loading buffer by boiling them for 10 min and then detection of the proteins by SDS-PAGE and silver staining. A pull-down assay using the GST tag alone was employed to identify proteins that nonspecifically bound to GST or the glutathione–Sepharose beads, which were used as the negative control. The GST-rMbovP0145-loaded glutathione–Sepharose beads that were not incubated with the cell supernatants were used as a negative blank control. After comparison with the control samples, differential protein bands bound to the GST-rMbovP0145-loaded glutathione–Sepharose beads were excised from the gel and identified using mass spectrometry (Applied Protein Technology Co. Ltd., Shanghai, China).

### 2.8. Immunoprecipitation Assay on Interaction between β-Actin and MbovP0145

The full length of the β-actin gene (GenBank accession no. NM_173979.3) was amplified from EBL cells and cloned into p3×Flag-CMV-10 to obtain recombinant plasmid p3×Flag-CMV-10-actin by restriction digestion with EcoR1 and BamH1. The Mbov_0145 gene was cloned into the pCAGGS-HA vector (kindly provided by Professor Shaobo Xiao from the College of Veterinary Medicine, Huazhong Agricultural University, Wuhan, China) by restriction digestion with EcoR1 and XhoI to generate the pCAGGS-HA-Mbov_0145 plasmid. The insertion in both plasmids was verified to be correct by sequencing. To further investigate the protein interactions between MbovP0145 and β-actin, HEK293T cells in a 10-cm dish were cultured at 37 °C in an incubator with 5% CO_2_ and transfected with both plasmids simultaneously with Lipofectamine 2000 DNA transfection reagent (Invitrogen, Carlsbad, CA, USA) according to the manufacturer’s instructions. After 36 h of incubation, the cells were harvested and lysed with NP-40 buffer containing 1 mM PMSF and 1 mg/mL cocktail (Roche, Mannheim, Germany) at 4 °C for 1 h. The lysates were pretreated with 50-μL protein A/G agarose beads (Beyotime Biotechnology) for 1 h at 4 °C, and then, the beads were removed by centrifugation. Next, 2 μg of anti-HA antibody (MBL, Nagoya, Japan) was added to these pretreated lysates and further incubated overnight at 4 °C. Protein A/G agarose beads were added to the mixture and incubated at 4 °C for 2 h on a rocking platform. The beads were collected by centrifugation, washed five times with NP-40 buffer, and resuspended in SDS loading buffer. These eluted proteins were subjected to a Western blot assay as mentioned above but used anti-Flag and anti-HA polyclonal antibodies (MBL) to probe the tagged proteins (3× Flag-actin and HA-MbovP0145).

### 2.9. Colocation Analysis with Confocal Laser Fluorescence Microscopy 

The colocation analysis was performed as described previously [19]. Briefly, EBL cells (5 × 10^4^/mL) were grown on glass coverslips using 12-well plates. The cells were incubated with 20-μg rMbovP0145 for 24 h at 37 °C. After washing with PBS, the cells were fixed with 4% formaldehyde solution for 15 min and permeabilized with 0.1% Triton X-100 for 10 min at room temperature, followed by blocking using 2% BSA in PBS for 1 h. Later, the cells were probed with mouse anti-rMbovP0145 polyclonal antibodies (dilution at 1:300) and rabbit anti-β-actin polyclonal antibodies (Cell Signaling Technology, Beverly, MA, USA; dilution at 1:100) in PBS at 4 °C overnight. After washing with PBS, the secondary antibodies Alexa Fluor 488-labeled goat anti-mouse IgG (Life Technologies, Thermo Fisher Scientific; dilution of 1:300) and Alexa Fluor 594-labeled goat anti-rabbit IgG (Life Technologies, Thermo Fisher Scientific; dilution of 1:300) were incubated with the cells for an additional 1 h. The cell nuclei were counterstained with DAPI (Beyotime Technology). Finally, the slides were cover-slipped and observed with a confocal laser fluorescence microscope (Olympus FV1000 and IX81, Tokyo, Japan). Furthermore, HEK293T cells transfected with pEGFP-C1-145 or the empty vector pEGFP-C1 were fixed with 4% formaldehyde solution for 15 min and permeabilized with 0.1% triton X-100 for 10 min. After washing and blocking, HEK293T cells were incubated with rabbit anti-β-actin antibody overnight at 4 °C. The cells were then washed and further incubated with Alexa Fluor 594-labeled goat anti-rabbit IgG, while the cell nuclei were labeled with DAPI.

### 2.10. Knockdown Assay of β-Actin with siRNA Interference 

Small interfering RNA (siRNAs) targeting β-actin (ACTB) and non-targeting control siRNA as the negative control (siCtrl) were designed and synthesized by Genepharam (Shanghai, China). The siRNA sequences were as follows: siACTB sense: GCAUUCACGAAACUACCUUTT, siACTB antisense: AAGGUAGUUUCGUGAAUGCTT, siCtrl sense: UUCUCCGAACGUGUCACGUTT, and siCtrl antisense: ACGUGACACGUUCGGAGAATT. EBL cells (5 × 10^4^/mL) grown in 12-well plates were transfected with three concentrations (50 nM, 80 nM, and 100 nM) of siRNA or siCtrl using jetPRIME transfection reagent according to the manufacturer’s instructions. The knockdown efficiencies were quantified by qRT-PCR and Western blot analysis. EBL cells with β-actin knockdown were used to evaluate the impact of β-actin on IL-8 mRNA expression and the activation of MAPK under different conditions.

### 2.11. Inhibition of Actin Polymerization by Cytochalasin D 

Cytochalasin D (Cyt D) (Sigma-Aldrich, St. Louis, MO, USA) was used as the inhibitor of actin polymerization. A CCK-8 assay was performed to evaluate the cytotoxicity of Cyt D to EBL cells (Dojindo, Shanghai, China), as previously described [11]. Briefly, 5 × 10^3^ cells/well were seeded and allowed to grow for 24 h in 96-well plates and then treated with Cyt D (0.1 μM, 0.5 μM, 1 μM, 2.5 μM, 5 μM, 10 μM, or 20 μM) for 24 h. Cells treated with DMSO were used as the negative control, and cells with no treatment were used as the mock group. Next, each well was incubated with 10 μL CCK-8 for 2 h. The light absorption value at 450 nm was detected by the microplate spectrophotometer (PerkinElmer Victor NIVO 3S, Waltham, MA, USA). Each treatment was carried out in three repeats, and all experiments were performed independently three times. After that, the monolayer EBL cells (2 × 10^5^ cells in a 12-well plate) were treated with the selected concentration of Cyt D for 2 h, followed by the addition of 8 μg/mL rMbovP0145 and incubation for more than 12 h. The total RNA was then extracted, and qRT-PCR was used to analyze the expression of IL-8 mRNA.

### 2.12. Analysis of the Molecules Critical to Signaling Pathways

Initially, EBL cells were treated with rMbovP0145 (8 μg/mL) for 15 min, 30 min, 60 min, or 120 min. Posttreatment, the cells were lysed in RIPA lysis buffer (Sigma-Aldrich) containing inhibitors of proteases and phosphatases (Roche, Basel, Switzerland) and then boiled in 5 × SDS loading buffer. The cell lysate was separated by 10% SDS-PAGE, followed by immunoblotting with the antibodies against ERK1/2, P38, phosphorylated ERK1/2, phosphorylated p38, and GAPDH (Cell Signaling Technology). To confirm the signaling pathways that contribute to IL-8 production, the EBL cells were pretreated with the inhibitors U0126 (Topscience, Shanghai, China) for ERK1/2 and SB203580 (Topscience) for P38 for 1 h. The cells were then stimulated with rMbovP0145, and the IL-8 mRNA levels were quantified as mentioned above.

### 2.13. Neutrophil Chemotaxis Assay

Neutrophils were isolated from the peripheral blood of healthy cattle by Ficoll-Hypaque gradient centrifugation as previously described [12]. Briefly, neutrophils were isolated immediately under sterile conditions with Ficoll (TBD Science, Tianjin, China). The mixed red blood cells were then removed with the Red Blood Cell Lysis Buffer (TBD Science), and the cells were resuspended in DMEM medium supplemented with 10% FBS.

A chemotaxis assay was carried out as described previously [20]. Briefly, neutrophils (1 × 10^5^/mL) were resuspended in 200 μL DMEM medium and added to the upper Transwell chamber (3-μm pores, Costar, Corning, NY, USA). Next, 600 μL culture supernatant from each group of EBL cells infected with *M. bovis* HB0801, T6.93 (*M. bovis* ΔMbov_0145), and its complementary strain CT6.93 for 24 h were centrifuged at 10,000× *g* for 5 min to collect the mycoplasma-free and cell-free supernatant. DMEM medium from the uninfected EBL cells served as the negative control. These culture supernatants were added to the lower wells. The Transwell plate was then incubated at 37 °C for 8 h. A hemocytometer was used to count the cells that migrated to the lower chamber. The Transwell migration assay was repeated three times independently.

### 2.14. Statistical Analysis

Data were expressed as the mean ± SD, and statistical significance was determined using a Student’s *t*-test. GraphPad Prism software (version 7.0) was used for statistical analysis, and significant differences were determined as * *p* < 0.05, ** *p* < 0.01, and *** *p* < 0.001.

## 3. Results

### 3.1. Bioinformatic Prediction of Genomic Features for the Mbov_0145 Gene

The Mbov_0145 gene is located between 162,942 nt and 164,387 nt within the genome of *M. bovis* HB0801. According to ProtParam (NCBI Protein ID no. AFM51518.1), MbovP0145 has a molecular weight of 55 kDa and an isoelectric point of 8.90. The average hydrophobicity of MbovP0145 is −0.558, with an instability index of 36.36. MbovP0145 contains 66 positively charged residues and 60 negatively charged residues, of which Lysine and Asparagine make up 10.8% and 10.0%, respectively. According to the PSORTb database, the location of the protein was found to be in the outer membrane. A putative lipoprotein signal peptide (Sec/SPII) was identified between residues 13 and 14 using SignalP. After searching the NCBI database for homologs of MbovP0145, two DUF285 domains were found between residues 233–378 aa and 334–453 aa, respectively. The putative DUF285 region has a moderate similarity with the BspA-like protein of the leucine-rich repeat (LRR) surface protein (Figure 1A). Overall, this protein belongs to Group 1 of the phylogenetic tree that is highly homologous to that of other *M. bovis* strains, including *M. bovis* 08M, *M. bovis* NingXia-1, and *M. bovis* MJ3. These have been annotated as BspA family LRR surface proteins, but there is a relatively long evolutionary distance from the reported bacteria in Group 2 (Figure 1B).

### 3.2. MbovP0145 Specifically Induced IL-8 Expression in EBL Cells

Using a Western blot assay with mouse polyclonal antibodies against rMbovP0145 as the probe and the abundant membrane protein MbovP0579 as the internal reference, the expression of MbovP0145 of about 55 kDa was confirmed in *M. bovis* HB0801 and CT6.93 but not in T6.93 (Figure 2A). Each strain was further characterized by comparing their proliferation under axenic medium and cell coculture conditions; there was no obvious difference in growth among the three strains (Appendix A). Next, the capacity of the three strains to induce IL-8 expression at the mRNA level in infected EBL cells was compared. Referring to HB0801-infected EBL cells at 6 h, 12 h, and 24 h post-infection (PI), the levels of IL-8 were significantly reduced in T6.93-infected EBL cells (*** *p* < 0.001); however, the levels in CT6.93-infected cells were increased and were restored to the levels in the WT-infected group at 12 h and 24 h PI (Figure 2B). Further, the expression kinetics of IL-8 in EBL cells treated with rMbovP0145 at six concentrations from 2 μg/mL to 20 μg/mL were investigated. The results showed that rMbovP0145 promoted IL-8 mRNA expression in a dose-dependent manner in EBL cells (Figure 2C). Similarly, the induction of IL-8 expression in EBL by rMbovP0145 was tested at different time intervals from 0 to 24 h to check its time-dependent expression (Figure 2D). These data indicated that MbovP0145 could specifically promote IL-8 expression in EBL cells during *M. bovis* infection. 

### 3.3. MbovP0145 Induction of IL-8 Expression Is Regulated by the MAPK Pathway

The phosphorylated P38 and ERK1/2 proteins were found to be upregulated in rMbovP0145-stimulated EBL cells compared to mock EBL cells (Figure 3A). The phosphorylation of P-P38 and P-ERK1/2 in EBL cells was dependent on their exposure time, with rMbovP0145 as indicated by a significant higher level of P-P38 and P-ERK1/2 during the early time (15 min) of exposure (Figure 3B,C, *** *p* < 0.001). Correspondingly, both inhibitors of P38 (SB203580) and ERK1/2 (U0126) significantly decreased rMbovP0145-induced IL-8 expression (Figure 3D, *** *p* < 0.001), which indicated that rMbovP0145-induced IL-8 expression is mediated by the P38 and P-ERK1/2 pathways.

### 3.4. Preliminarily Identification of MbovP0145 Interactive Proteins

The cellular proteins that potentially interact with MbovP0145 of *M. bovis* were identified by a GST pulldown assay using recombinant proteins GST-0145 and the GST empty vector protein (Appendix A). Both the GST and GST-0145 pulldowns detected several common bands in the infected EBL cells. However, two different bands generated by silver staining were identified in the GST-0145 protein lane (Figure 4A). These two specific bands (A and B) were then excised and subjected to a mass spectrometry analysis, and the cellular proteins that could possibly interact with the MbovP0145 protein were preliminarily identified to be actin and myosin (Table 2). β-actin of 42 kDa was used for further confirmation of its interaction with MbovP0145. 

### 3.5. Confirmation of β-Actin as the Interactive Protein of MbovP0145

The WCLs of HEK293T co-transfected with the plasmids expressing 3× Flag-tagged β-actin and HA-tagged MbovP0145, and the cells co-transfected with the plasmids only expressing tags as a negative control were subjected to a Western blot assay with either anti-HA mAb or anti-Flag mAb. The results showed that the tagged proteins were successfully expressed (Figure 4B, lower panel). The interaction between MbovP0145 and β-actin was verified with a coimmunoprecipitation (CO-IP) assay using the above cell lysates. As a result, Co-IP with an anti-HA mAb and protein A/G beads generated the complex of HA-MbovP0145 and 3× Flag-actin but not with 3× Flag (Figure 4B, upper panel). 

Further, the interaction between MbovP0145 and β-actin was confirmed by a colocalization assay. Double-immunofluorescence staining of EBL cells treated with 20 μg rMbovP0145 for 24 h showed that the colocalization of MbovP0145 and β-actin was distributed in the cell membrane and cytoplasmic parts (Figure 4C). To confirm if endogenous MbovP0145 also colocalized with β-actin in the EBL cells, the endotoxin-free plasmid of pEGFP-CI-145 was transfected with HEK293T cells, and the green fluorescence-labeled MbovP0145 was clearly colocalized with the red fluorescence-labeled β-actin shown by the yellow color of the merged images. In pEGFP-CI-transfected cells, there was no colocalization between MbovP0145 and β-actin (Figure 4D). Taken together, MbovP0145 can interact specifically with β-actin in the cell membrane and cytoplasm.

### 3.6. IL-8 Expression Induced by MbovP0145 Depends on Its Interactive β-Actin 

The involvement of β-actin in IL-8 expression induced by rMbovP0145 was evaluated by the transient transfection of EBL cells with three different concentrations of siRNA specifically targeting β-actin. The knockdown efficiency of the siACTB interference group was confirmed to be high by qRT-PCR and Western blot analysis compared to the control (Figure 5A,B). Next, 100 nM siACTB or siCtrl transfected cells were treated with or without rMbovP0145, and the IL-8 expression was measured. As expected, the IL-8 mRNA levels were upregulated upon rMbovP145 treatment alone (Figure 5C, *** *p* < 0.001), and the suppression of β-actin with 100-nM siACTB significantly decreased the IL-8 mRNA level by 1.6-fold induced by MbovP0145 (Figure 5C, *** *p* < 0.001). Collectively, β-actin could be specifically and positively be involved in IL-8 expression induced by rMbovP0145 in EBL cells. Subsequently, the knockdown of β-actin expression with siACTB prior to rMbovP145 stimulation significantly reduced the expression of P-P38 and P-ERK1/2 in EBL cells compared with the treatment with siCtrl (Figure 5D–F, *** *p* < 0.001), suggesting that β-actin might be an important bridge to activate these signal pathways.

The use of Cyt D, a specific inhibitor of actin polymerization, has demonstrated that actin depolymerization may play an important role in the MboP0145-induced IL-8 expression and MAPK phosphorylation levels in EBL cells. First, the optimum concentration of Cyt D determined by a CCK 8 viability assay was determined, and it was found to be 1-μM Cyt D, because it was the highest concentration that did not significantly affect the cell viability compared with the control cells (Figure 6A). Next, it was found that a pretreatment of EBL cells with 1 μM Cyt D could significantly enhance the IL-8 expression by 12.7-fold in cells responding to rMbovP0145 stimulation (Figure 6B, *** *p* < 0.001). In addition, the depolymerization of β-actin with Cyt D before rMbovP0145 treatment enhanced the expression of P-P38 and P-ERK1/2 activated by rMbovP0145 (Figure 6C–E).

The effect of Cyt D and siACTB on the interaction between rMboP0145 and β-actin was assessed via fluorescence confocal microscopy of EBL cells. The cells in the control group displayed abundant red fluorescence (Figure 6F, control). After the treatment of Cyt D prior to rMbovP0145 stimulation, the average fluorescence intensity of labeled β-actin did not change but decreased in the siACTB group, although the difference was not significant (Figure 6G).

### 3.7. MbovP0145 Induces Neutrophil Migration by Regulating the Production of IL-8

Based on the above findings, we used the Transwell assay to evaluate the chemotactic effect of the conditioned medium supernatant from the culture of *M. bovis*-infected cells on the chemotaxis of neutrophils. As seen in Figure 7, the chemotactic levels of neutrophils in the *M. bovis* infection groups were significantly higher than those of the negative control groups (medium and EBL medium). Among the groups infected with the three *M. bovis* strains, compared with *M. bovis* HB0801 and CT6.93, the conditioned medium from the mutant T6.93-infected EBL cells had a significantly lower effect on the chemotaxis of the neutrophils (* *p* < 0.05), while the chemotaxis effect of CT6.93 was restored to the level of HB0801 (ns *p* > 0.05). These results indicated that MbovP0145 plays an important role in neutrophil migration during *M. bovis* infections.

## 4. Discussion

Secreted proteins of mycoplasmas might function as virulence-related factors, it is difficult to confirm them due to the limited and inefficient genetic tools and cell/animal models [5]. We previously identified MbovP0145 as a unique secreted protein of *M. bovis* with Sec/SPII signal peptides and determined that it is worth to investigate further, because it has two DUF285 domains and BspA, the family of LRR surface proteins. The DUF285 domain appears distantly related to the LRRs [21]. Meanwhile, BspA was found to bind with the extracellular matrix component i.e., fibronectin and the clotting factor fibrinogen, which promotes bacterial adhesion and invasion [22,23]. In addition, we were able to explore the function of MbovP0145, because we fortunately obtained the mutant T6.93 (*M. bovis* ΔMbov_0145) without expression of MbovP0145 from our *M. bovis* mutant library [8] and constructed the complementary strain CT6.93 of this mutant. By following the hint of these domains’ functions in other bacteria, this study demonstrated that MbovP0145 could promote upregulation of the IL-8 expression in EBL through activation of the MAPK signal pathway, which might contribute to neutrophil migration. Further, we determined that β-actin is the interactive ligand of MbovP0145 (Figure 8).

As reported previously, the proteins with a DUF285 domain and LRR can promote bacterial adhesion and invasion of epithelial cells [23] and trigger the release of proinflammatory cytokines from monocytes and chemokine IL-8 from gingival epithelial cells by activating the TLR2-dependent pathway in vitro [24] and in vivo [25] in *T. forsythia* and the induction of chemotaxis in *Entamoeba histolytica* [13]. Among the proinflammatory cytokines, IL-8 is closely related to mycoplasmal infection. During *M. bovis* infection, the mRNA level of IL-8 and other inflammatory cytokines like IL-1β, IL-6, TNF-a, MMP-3, and MMP-8 were upregulated in bovine mammary epithelial cells and synovial cells [26,27]. Among them, IL-8 as a response of neutrophils to *M. bovis* has been previously reported by the migration and activation of neutrophils from the blood to the site of infection during *M. bovis* infection [28]. It has been reported that *M. pneumoniae* components (WCLs and lipid-associated membrane proteins) could induce IL-8 production in bronchial epithelial cells [29,30]. In addition, normal human bronchial epithelial (NHBE) cells infected with live *M. pneumoniae* increased IL-8 production, which was associated with the secretion of CARDS toxin. In turn, IL-8 stimulated the activity of neutrophils [31,32]. In agreement with a previous study, we found that *M. bovis*-enhanced IL-8 expression is mediated by MbovP0145. 

Neutrophil infiltration is the characteristic pathological feature of mycoplasma infection, and the IL-8/neutrophil axis plays an important role in the pathogenesis of mycoplasma [32]. IL-8 is a potent chemoattractant and activator of neutrophils, which is closely related to the occurrence and maintenance of inflammation [33]. IL-8 can be produced by a variety of cell types, including epithelial cells, macrophages, and neutrophils [34]. We conducted a neutrophil migration cell model based on a Transwell experiment in vitro and found that the migration of neutrophils infected with the Mbov_0145 mutant T6.93 was significantly decreased compared with the WT and the complementary strain CT6.93, indicating that MbovP145 was associated with the recruitment and activation of neutrophils. Although MbovP145-induced IL-8 might likely contribute to neutrophil migration, other cytokines could also be involved in this process, something that requires further investigation.

As is well-known, β-actin forms basal microfilament bundles, which is important for the structure and function of adhesion junctions in epithelial cells, and determines an apical-basal cell polarity [35]. The suppression of β-actin causes the loss of intercellular contact between the epithelial cells. In addition, β-actin is a major cytoskeletal protein and mediator of internal cell motility [36] and is highly conserved and involved in many cellular processes, including cell adhesion, cell migration, cytokinesis, endocytosis, cell division, and signal transduction [37]. Recently, it has been reported that extracellular actin in swine lungs is an important receptor for *M. hyopneumoniae* colonization [38]. In sepsis, β-actin can interact with TREM-1 on platelets to enhance the inflammatory response [17]. Moreover, several cytoskeleton-disrupting agents can affect some inflammatory signal transduction pathways [39,40]. In this study, we confirmed the interaction between the β-actin of EBL cells and MbovP0145 using pull-down assays, Co-IP, and confocal microscopy observations of their colocalizations. Furthermore, to explore whether MbovP145 interactions with β-actin can affect IL-8 production or not, we used two different approaches to inhibit β-actin expression: the transfection of siRNA of ACTB and treatment of EBL cells with the actin-depolymerizing drug Cyt D. The data revealed that β-actin suppression reduced the levels of IL-8 expression and phosphorylation of the ERK1/2 and P38 molecules, critical to the MbovP0145 stimulation-associated signaling pathways. However, different with siACTB treatment, Cyt D treatment increased the levels of IL-8 expression and phosphorylation of ERK1/2 and P38 in EBL cells stimulated by MbovP0145. These findings are consistent with a previous study in which Cyt D increased IL-8 expression, which induced the reorganization of the actin cytoskeleton and the rapid activation of signal transduction [41]. In addition, actin depolymerization can induce NF-κB activation and subsequent inflammatory mediators [42]. Therefore, β-actin in the form of depolymerization might be an interactive ligand of MbovP145 to activate MAPK signaling to produce IL-8. In contrast, beta1 integrin signaling is a common pathway of IL-8 production of epithelial cells. Some of them are also receptors for bacteria invasions. It has been reported that α5β1 integrin ligation on an extracellular matrix fibronectin fragment in human PMNLs inhibited chemotaxis toward IL-8 and then mediated MAPK signaling [43]. Interestingly, the linkage between actin and integrins plays a key role in actin polymerization and organization to promote adhesion and other signal pathways [44]. Combined with our results in this study, we identified that actin could be one of MbovP0145-binding proteins and whether the actin-binding protein connects with Beta1 integrins is worth further investigation. 

Moreover, the Myosin heavy-chain 9 around 200 kDa might also interact with MbovP145, because it was co-identified by the pull-down assay. We speculate that it would be possible, because it has both ATP- and actin-binding sites and might work together with β-actin and MbovP0145 in cell adhesion, migration, proliferation, and differentiation [45]. Therefore, it is worth being studied in the future. Of course, it would be more difficult to confirm its function due to its large size.

Some limitations in this study should be discussed. First, we detected IL-8 expression on the transcriptional level after stimulation with rMbovP0145 in EBL cells. The protein level of IL-8 expression would be preferred in future studies to support these solid conclusions. However, the IL-8 expression by ELISA and the Western blot assay was too low to detect (data not shown). We speculate less IL-8 was secreted, probably because of their limitation in EBL cells. Further studies will be needed to prove the phenomenon using other bovine primary cells in vitro. 

## 5. Conclusions

The present study demonstrated that MbovP0145, a secreted protein of *M. bovis*, interacts with β-actin to enhance IL-8 expression in EBL cells through MAPK activation and contributes to the recruitment and activation of neutrophils during infection. These findings increase the understanding of the pathogenesis of *M. bovis* and innate immunity induced by this pathogen. 

## Figures and Tables

**Figure 1 pathogens-10-01628-f001:**
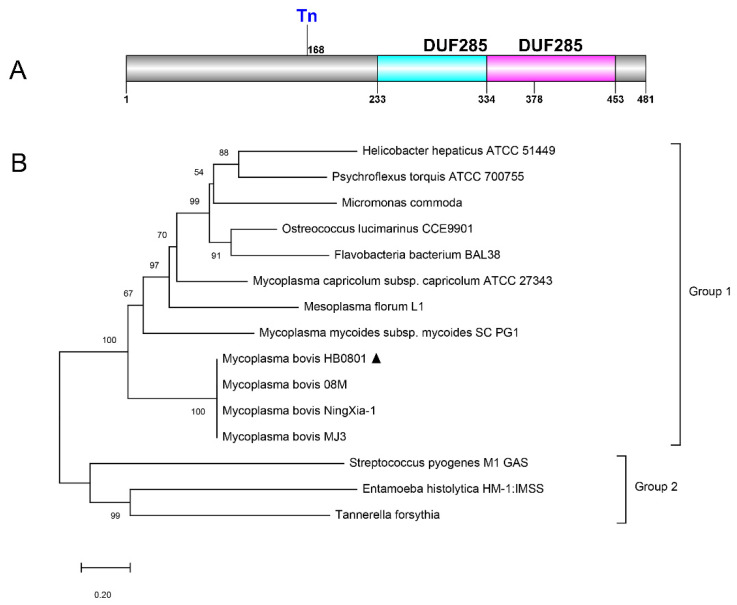
Overall gene organization of Mbov_0145 and its phylogenetic analysis. (**A**) Schematic representation of the domain structure of MbovP0145 via the NCBI database. (**B**) Phylogeny of MbovP0145 and orthologs from other related bacteria. The trees were constructed using the neighbor-joining method with nodal support assessed by 1000 bootstrap replicates. Bacteria from which complete SPase I sequences were taken and their respective GenBank or NCBI reference sequence accession numbers are as follows: *M. bovis* HB0801 (AFM51518.1), *M. bovis* 08M (AQU85457.1), *M. bovis* NingXia-1 (ATQ40684.1), *M. bovis* MJ3 (AXJ69624.1), *Mycoplasma mycoides subsp. mycoides* SC PG1 (WP_011167164.1), *Mycoplasma capricolum* subsp. *capricolum* ATCC 27343 (WP_011387196.1), *Mesoplasma florum* L1 (WP_011183344.1), *Ostreococcus lucimarinus* CCE9901 (XP_001422294.1), *Helicobacter hepaticus* ATCC 51449 (WP_011114893.1), *Psychroflexus torquis* ATCC 700755 (ZP_01255287.1), *Micromonas commode* (ACO63939.1), *Flavobacteria bacterium* BAL38 (ZP_01734433.1), *Streptococcus pyogenes* M1 GAS (AAK33772.1), *Entamoeba histolytica* HM-1:IMSS (XP_653526.1), and *Tannerella forsythia* (WP_157755308.1). *M. bovis* HB0801 is indicated by the triangle marker “▲”.

**Figure 2 pathogens-10-01628-f002:**
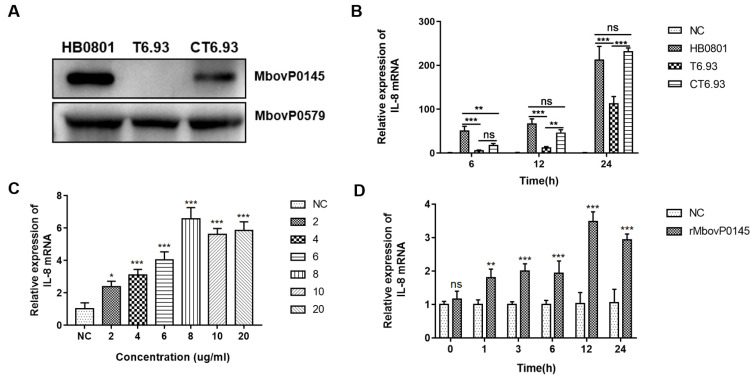
Role of MbovP0145 in IL-8 expression in EBL cells. (**A**) The expression of MbovP0145 in the *M. bovis* HB0801, mutant strain T6.93, and complemented strain CT6.93, respectively. Total cell lysates of each strain were separated on 10% SDS-PAGE gel, transferred to a polyvinylidene fluoride membrane, and treated with anti-MbovP0145 and anti-MbovP579 antibodies. (**B**) *M. bovis* promotes IL-8 mRNA expression in EBL cells. EBL cells were infected with *M. bovis* HB0801, T6.93, or CT6.93. Cells were collected at different time points, and qRT-PCR was performed to quantify the IL-8 expression. PBS was used as a negative control. (**C**,**D**) EBL was stimulated by rMbovP0145 at different concentrations and times. The relative expression of IL-8 was analyzed as above. Data are the means of three independent assays. Standard deviations are indicated by error bars. *p*-values are indicated by asterisks (*** *p* < 0.001, ** *p* < 0.01, * *p* < 0.05, and ns = *p* > 0.05).

**Figure 3 pathogens-10-01628-f003:**
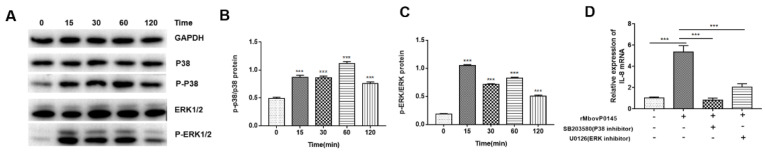
MbovP0145 induction of IL-8 expression is regulated by the MAPK pathway. (**A**) Effect of rMbovP0145 on P38 and ERK1/2 phosphorylation. Immunoblot analysis of phosphorylated p38, ERK1/2, and total GAPDH in EBL cells after rMbovP0145 stimulation. The cells were stimulated with 8 μg/mL rMbovP0145 for 0 min-120 min. (**B**,**C**) Densitometry quantification of immunoblot analysis results of phosphorylated P38 and ERK1/2 presented relative to those of basal P38 and ERK1/2. GAPDH was used as a loading control. (**D**) Effect of P38 and ERK1/2 inhibition on the production of IL-8 induced by rMbovP0145. EBL cells were treated with (+) or without (−) 1 μM SB203580 and 1 μM U0126 for 1 h prior to rMbovP0145 treatment. The cells were collected at 12 h, and the mRNA level of IL-8 was determined. Data are the means of three independent assays. Standard deviations are indicated by error bars. *p*-values are indicated by asterisks (*** *p* < 0.001).

**Figure 4 pathogens-10-01628-f004:**
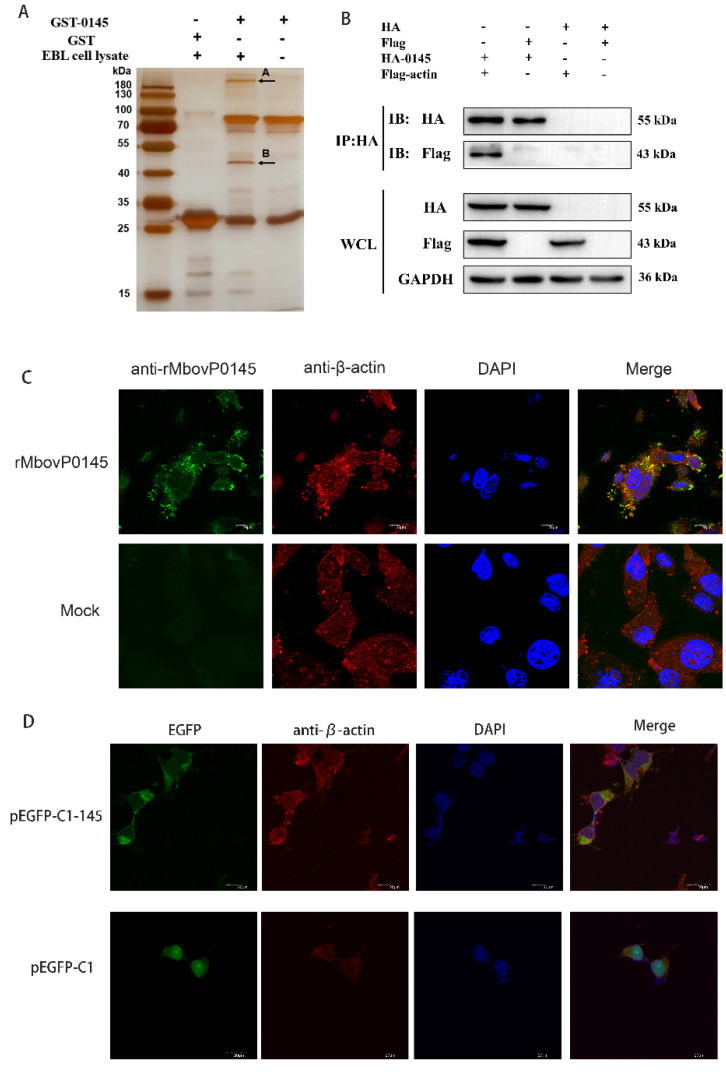
Identification of the MbovP0145-interacting protein by GST pulldown and confirmation of β-actin interaction with MbovP0145. (**A**) MbovP0145-interacting protein by GST pulldown. Two differential protein bands (indicated by (**A**,**B**)) were discovered by comparison with the control samples. Two bands were excised from the gel and identified by MS. (**B**) Coimmunoprecipitation of MbovP0145 with β-actin. HEK293T cells were co-transfected with 3× Flag-actin and HA-MbovP0145, and the whole-cell lysates obtained at 36 h post-transfection were immunoprecipitated (IP) with anti-HA mAb. After separation by SDS-PAGE, MbovP0145 and β-actin were then detected with Western blotting assays using antibodies against either the HA or Flag tag. The identities of the protein bands are indicated on the right. (**C**) Colocalization of the exogenous rMbovP0145 protein (green) and β-actin (red). EBL cells were treated with the rMbovP0145 protein for 24 h. Cells were fixed and subjected to indirect immunofluorescence to detect MbovP0145 (green) and β-actin (red) with mouse anti-MbovP0145 and rabbit anti-β-actin antibodies, respectively. The position of the nucleus is indicated by DAPI (blue) staining in the merged image. (**D**) Colocalization indicated that pEGFP-C1-MbovP145 transfection could co-locate with β-actin in HEK293T cells. The scale bars in the figure represent 20 μm.

**Figure 5 pathogens-10-01628-f005:**
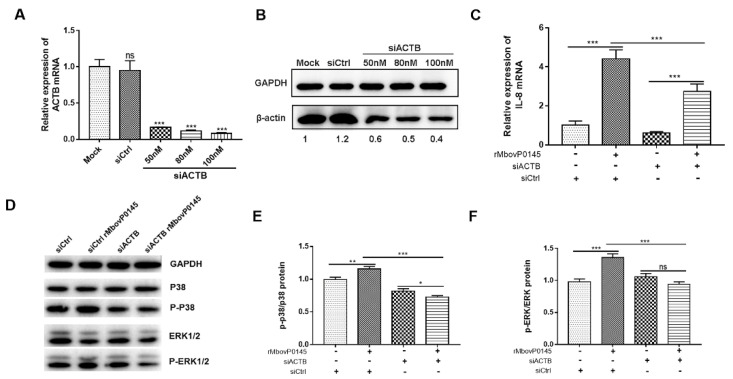
IL-8 expression and MAPK phosphorylation levels in EBL cells after β-actin suppression with siACTB. (**A**) Specific siRNA with different concentrations could significantly suppress β-actin expression compared with mock EBL cells at the mRNA level. (**B**) Specific siRNA suppressed β-actin expression compared with mock group at the protein level. (**C**) siRNA-mediated β-actin interference suppressed rMbovP0145-induced IL-8 mRNA expression at 12 h. (**D**) Effects of β-actin knockdown on the activation of the MAPK signaling pathway. EBL cells were transfected with 100-nM siCtrl or siACTB for 24 h and then treated with rMbovP0145 as above. The total protein was extracted and detected using Western blotting. (**E**,**F**) The intensity of the immunoblots was evaluated with ImageJ, and the fold changes of P-P38/P38 and P-ERK1/2/ERK1/2 are shown. All data are the means of three independent assays. Standard deviations are indicated by error bars. *p*-values are indicated by asterisks (*** *p* < 0.001, ** *p* < 0.01, * *p* < 0.05, and ns = *p* > 0.05).

**Figure 6 pathogens-10-01628-f006:**
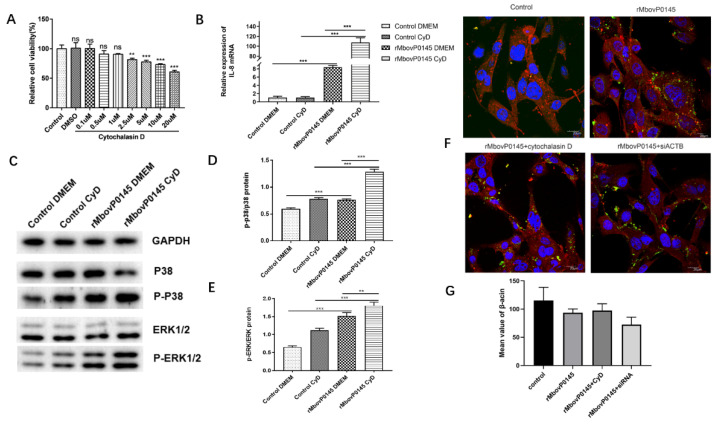
IL-8 expression and MAPK phosphorylation levels in EBL cells after cytochalasin D treatment. (**A**) Cell cytotoxicity of cytochalasin D. EBL cells (5 × 10^3^ per well) were treated with different concentrations of cytochalasin D for 24 h, and CCK-8 was used to detect the cell viability. Wells treated with DMSO were used as the negative control. (**B**) Cytochalasin D significantly enhanced the mRNA expression of IL-8. EBL cells were pretreated with 1 μM cytochalasin D or PBS for 2 h, and then, the cells were stimulated with 8 μg/mL rMbovP0145 protein for 12 h. The mRNA level of IL-8 was analyzed as above. (**C**) Effects of β-actin depolymerize on the MAPK signaling pathway. EBL cells were pretreated with cytochalasin D (1 μM) or PBS for 2 h and then treated with rMbovP0145 as above. The total protein was extracted and detected using Western blotting. (**D**,**E**) ImageJ was used to detect the gray degree values. All data are the means of three independent assays. Standard deviations are indicated by error bars. *p*-values are indicated by asterisks (** *p* < 0.01 and *** *p* < 0.001). (**F**) Fluorescence confocal microscopic analysis of β-actin in EBL cells. The cells were treated with 1 μM cytochalasin D or 100 nM siACTB prior to rMbovP0145 stimulated. β-actin was stained with Alexa Fluor 564-conjugated goat anti mouse IgG antibody (red). rMbovP0145 was hybridized with mouse anti-rMbovP0145 polyclonal antibody and labeled with FITC-conjugated goat anti mouse IgG antibody green fluorescence (green). The nuclei were counterstained with DAPI (blue). All the cell samples were examined with a laser confocal scanning microscope. Scale bar = 20 μm. (**G**) Acquired confocal microscopy images were analyzed for Alexa Fluor 564 channel intensity representing β-actin staining using ImageJ software.

**Figure 7 pathogens-10-01628-f007:**
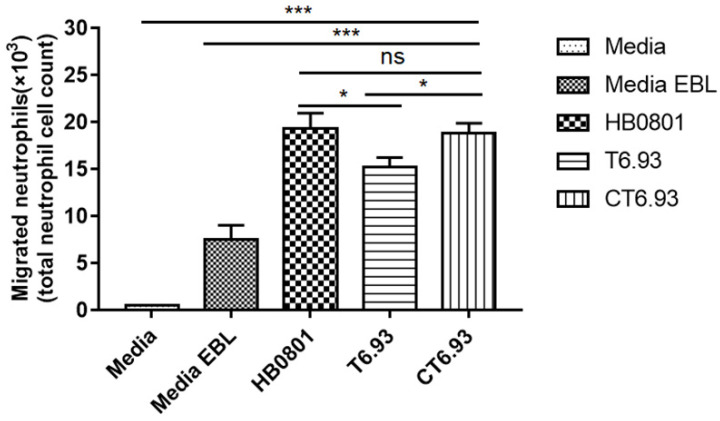
The mutant strain of MbovP0145 impaired the ability of neutrophil recruitment. Neutrophils were assayed for chemotaxis using a conditioned medium collected after the exposure of EBL cells to *M. bovis* (HB0801), the Mbov_0145 mutant strain (T6.93), and the complementary strain (CT6.93). DMEM or DMEM from uninfected EBL cells served as the negative control. After being cocultured 8 h, the neutrophils that migrated to the lower chamber were collected and counted. Data are the means of three independent assays. Standard deviations are indicated by error bars. *p*-values are indicated by asterisks (*** *p* < 0.001, * *p* < 0.05, and ns = *p* > 0.05).

**Figure 8 pathogens-10-01628-f008:**
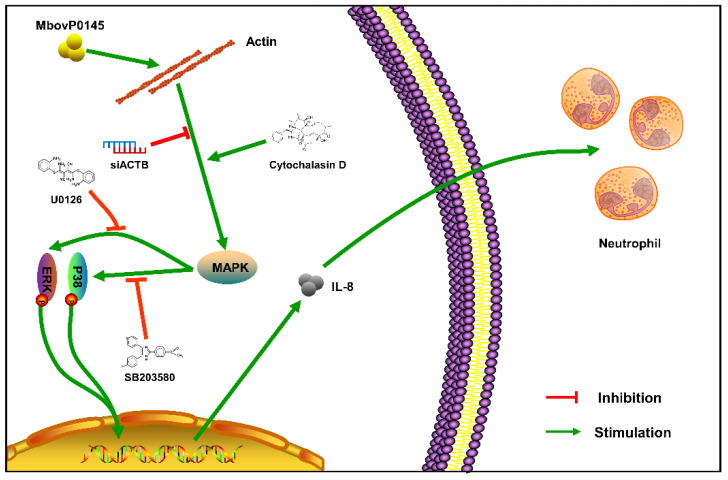
Diagram of the possible signaling pathway of MbovP0145-induced IL-8 expression. MbovP0145 interacted with β-actin in the cytoplasm and contributed to activating the MAPK-signaling pathway, subsequently upregulating the expression of IL-8 and facilitating neutrophile cell chemotaxis.

**Table 1 pathogens-10-01628-t001:** Oligonucleotide primers used for PCR or qPCR in this study.

Names	Primer Sequences (5′→3′) ^1^
IL-8-F	GAAGAGAGCTGAGAAGCAAGATCC
IL-8-R	ACCCACACAGAACATGAGGC
GAPDH-F	TGGTGAAGGTCGGAGTGAAC
GAPDH-R	ATGGCGACGATGTCCACTTT
pOH/P-0145-F1	ATTTGCGGCCGCACGGGGCTAAAGAAGCTGATAT (NotI)
pOH/P-0145-R1	TAGCAAAAAGCATAATTATTTATATCCTTTTCTT
pOH/P-0145-F2	TAAATAATTATGCTTTTTGCTAGTTCACTTCCTTT
pOH/P-0145-R2	AATTGCGGCCGCTTATTTAGATACTTGCCTAAAA (NotI)
GST-0145-F	TTCCAGGGGCCCCTGGGATCCATGCTGTTTGCCTCAAGCCTG (BamHI)
GST-0145-R	GTCACGATGCGGCCGCTCGAGTTATTTGCTAACCTGACGAAAATTCG (XhoI)
HA-0145-F	GTTCCAGATTACGCTGAATTCATGCTGTTTGCCTCAAGCCTG (EcoRI)
HA-0145-R	ATTAAGATCTGCTAGCTCGAGTTATTTGCTAACCTGACGAAAATTCG (XhoI)
Flag-actin-F	AAGCTTGCGGCCGCGAATTCAATGGATGATGATATTGCTGCGC (EcoRI)
Flag-actin-R	CAGGGATGCCACCCGGGATCCCTAGAAGCATTTGCGGTGGAC (BamHI)
EGFP-0145 F	AGTCCGGACTCAGATCTCGAGATATGCTGTTTGCCTCAAGCCTG (XhoI)
EGFP-0145 R	TTATCTAGATCCGGTGGATCCTTATTTGCTAACCTGACGAAAATTCG (BamHI)

^1^ Restriction enzyme sites are underlined.

**Table 2 pathogens-10-01628-t002:** LC-MS/MS analysis of the protein band extracted from the SDS-PAGE gel.

Band	Protein Name	Accession No.^a^	Unique Peptide Count ^b^	Percentage Coverage ^c^	MW (KDa) ^d^	PI ^e^
A	Myosin heavy chain 9	F1MQ37	92	35.62%	227,201.32	5.49
B	Actin, cytoplasmic 1	P60712	10	24.00%	41,736.29	5.29

^a^ Accession number in the UniProtKB database. ^b^ Unique peptide count means the more unique peptides, the higher the credibility of the protein. ^c^ Number of detected amino acids/total number of amino acids in the proteins. ^d^ Theoretical molecular weight of the proteins. ^e^ Isoelectric points of the proteins.

## Data Availability

Data sharing not applicable.

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
