# Peer review of "Secreted MbovP0145 Promotes IL-8 Expression through Its Interactive β-Actin and MAPK Activation and Contributes to Neutrophil Migration"

_pathogens, 2021, doi:10.3390/pathogens10121628_

Round 1
Reviewer 1 Report
The submitted manuscript presents a work with great relevance and interest to the scientific community.
The work is well outlined and the figures show care and accuracy in presentation.
The material and methods section is clear and concise, however it would be important to add more information regarding the production and purification of recombinant proteins (section 2.3 Cloning, expression, and purification of a recombinant MbovP0145 protein).
In section 2.8 (GST pull-down assay to identify interactive protein of MbovP0145) it is also important to specify the elution method (line 170)
The conclusions are clear and supported by the results presented.
Author Response
Point 1: The material and methods section is clear and concise, however it would be important to add more information regarding the production and purification of recombinant proteins (section 2.3 Cloning, expression, and purification of a recombinant MbovP0145 protein).
Response 1: Thank you for your critical comment. We have added the production information like E. coli BL21 competent cells and the concentration of IPTG when expression and purification of a recombinant MbovP0145 protein. Please see line 198-200.
Point 2: In section 2.8 (GST pull-down assay to identify interactive protein of MbovP0145) it is also important to specify the elution method (line 170)
Response 2: Thank you for your comment. We are sorry for our unclear description. The sentence was modified as follows and revised in our manuscript (line 276-279).
The beads were then washed four times with ice cold NP40 buffer by a rocking-turn blending, followed by elution with SDS loading buffer by boiling them for 10 min and then detection of the proteins by SDS-PAGE and silver staining.

Reviewer 2 Report
The authors characterized the secreted protein Mbov_0145 with respect to its ability to induce IL-8 secretion. This is a thorough work, which is novel and interesting to the readership of the journal. A set of informative experiments is laid out that enabled the authors to characterize the interactions of the protein Mbov_0145 with EBL cells. An analysis about the occurrence and conservation of the Mbov_0145 in other M. bovis strains would strenghten the manuscript and should be added. Therefore, genome sequences deposited in GenBank should be used. The discussion section needs to be expanded in the light current knowledge of IL-8 secretion of bovine tissue in response to M. bovis. The following papers describing the secretion of IL8 from bovine cells after they have been in contact with M. bovis should be discussed.
- Innate immune response in bovine neutrophils stimulated with Mycoplasma bovis, Veterinary Research, 2021, PMID: 33431726
- Inflammatory cytokine mRNA and protein levels in the synovial fluid of Mycoplasma arthritis calves, J Vet Med Sci, 2021, PMID: 33431726
- Effect of Mycoplasma bovis on expression of inflammatory cytokines and matrix metalloproteinases mRNA in bovine synovial cells, Vet Immunol Immunopathol, 2019, PMID: 31446205.
- The immune response of bovine mammary epithelial cells to live or heat-inactivated Mycoplasma bovis, PMID: 26211967
The canonical pathway of IL-8 production of epithelial cells does not involve beta actin but beta1 integrins signalling and should be discussed in the discussion section. The limitations of the study should be discussed (use of EBL cells, IL-8 detection via mRNA instead of antibodies etc.). A native English speaker should type-edit the written English before resubmission.
Minor comments:
Line 21-22: Replace “endotoxin-removed recombinant” with “recombinant”, and replace “could stimulate” with “stimulated”.
Line 54: Replace “affecting” with “driving”
Line 54-56: Rephrase since the sentence is not clear as it stands.
Line 68-69: I disagree that this study aimed to confirm a virulence factor, the sentence needs to be rephrased.
M&M section
The number of replicates (biological and technical) for the different assays carried out should be added.
Line 103-107: I suggest to call it “codon adaptation for expression in E. coli” instead of “mutagenesis”
Line 120: “In trans complementation of M. bovis T6.93 via a plasmid carrying the gene Mbov_0145” should be used as subheading
Line 151: What was the positive control used?
Line 152-161: Direct detection of the IL-8 would be more meaningful than mRNA expression especially in the absence of a positive control.
Line 162-178: A rGST tag was used as negative control according to the results section. Therefore, this control should be mentioned in the paragraph.
Line 265: the pores have a size of 3 µm NOT 3 mm as stated in the manuscript!
Author Response
Dear reviewers,
Thank you very much for your comments. In according with the comments of you and the reviewers, we have made extensive modification on the original manuscript and added extra data as required. Revised parts are highlighted in red in the paper. You will find the point-by-point responses to comments/questions as attached. We think the revised version was improved greatly and hope it will meet your requirement.

This manuscript is a resubmission of an earlier submission. The following is a list of the peer review reports and author responses from that submission.